# Nitric Oxide Is Involved in the Regulation of the Ascorbate–Glutathione Cycle Induced by the Appropriate Ammonium: Nitrate to Mitigate Low Light Stress in *Brassica pekinensis*

**DOI:** 10.3390/plants8110489

**Published:** 2019-11-11

**Authors:** Linli Hu, Yutong Li, Yue Wu, Jian Lv, Mohammed Mujitaba Dawuda, Zhongqi Tang, Weibiao Liao, Alejandro Calderón-Urrea, Jianming Xie, Jihua Yu

**Affiliations:** 1Gansu Provincial Key Laboratory of Aridland Crop Science, Gansu Agricultural University, Lanzhou 730070, China; hull@gsau.edu.cn; 2College of Horticulture, Gansu Agricultural University, Lanzhou 730070, China; liyutong_115@163.com (Y.L.); wuyue_gsau@163.com (Y.W.); lvjian@gsau.edu.cn (J.L.); mmdawuda@yahoo.com (M.M.D.); 184481739@163.com (Z.T.); liaowb@gsau.edu.cn (W.L.); xiejianming@gsau.edu.cn (J.X.); 3Department of Horticulture, FoA, University for Development Studies, P. O. Box, Tamale TL 1882, Ghana; 4Department of Biology, College of Science and Mathematics, California State University, Fresno, CA 97340, USA; calalea@csufresno.edu

**Keywords:** low light intensity, nitric oxide, ascorbate–glutathione cycle, ammonium: nitrate ratio, mini Chinese cabbage

## Abstract

Low light intensity is common in northern China due to fog or haze, and causes stress for crop plants. To solve the problem of low light intensity stress on the growth and development of vegetable crops in China, new cropping strategies must be developed. We previously showed that an appropriate ratio of ammonium and nitrate (NH_4_^+^:NO_3_^−^) can alleviate the effect of low light stress on plants, although it is not clear what mechanism is involved in this alleviation. We propose the hypothesis that an appropriate ammonium/nitrate ratio (10:90) can induce NO synthesis to regulate the AsA-GSH cycle in mini Chinese cabbage seedlings under low light intensity. To test the hypothesis, we conducted a series of hydroponic experiments. The results indicated that, under low light intensity conditions, appropriate NH_4_^+^:NO_3_^−^ (N, NH_4_^+^:NO_3_^−^ = 10:90) decreased the contents of malondialdehyde (MDA), hydrogen peroxide (H_2_O_2_), and superoxide anion (O_2_^−^) in leaves compared with nitrate treatment. Exogenous nitric oxide (SNP) had the same effects on MDA, H_2_O_2_, and O_2_^−^. However, with the addition of a NO scavenger (hemoglobin, Hb) and NO inhibitors (N-nitro-l-arginine methyl ester, L-NAME), NaN_3_ (NR inhibitor) significantly increased the contents of MDA, H_2_O_2_, and O_2_^-^. The application of N solution enhanced the AsA-GSH cycle by increasing the activities of ascorbate peroxidase (APX), glutathione reductase (GR), monodehydroascorbate reductase (MDHAR), dehydroascorbate reductase (DHAR), and ascorbate oxidase (AAO), compared with control (NH_4_^+^:NO_3_^−^ = 0:100). Meanwhile, exogenous SNP significantly increased the above indicators. All these effects of N on AsA-GSH cycle were inhibited by the addition of Hb, L-NAME and NaN_3_ in N solution. The results also revealed that the N and SNP treatments upregulated the relative expression level of *GR, MDHAR1, APXT, DHAR2*, and* AAO *gene in mini Chinese cabbage leaves under low light stress. These results demonstrated that the appropriate NH_4_^+^:NO_3_^−^ (10:90) induced NO synthesis which regulates the AsA-GSH cycle in mini Chinese cabbage seedlings under low light stress.

## 1. Introduction

Low light intensity is common in northern China due to fog or haze, and causes a stress to crop plants [1]. In addition, it has been reported that a significant reduction in solar radiation reaching the earth has occurred during the last 50 years [2]. Low light environment usually reduces plant size, plant height, and stem thickness, and may also affect energy distribution in specific parts of the plant as well as the reproductive capacity of plants [3]. It has been reported that net photosynthesis in plants usually decreases under low light intensity [4,5]. Ding et al. [6] reported that under low light stress, soluble protein content, and catalase activity decreased; however, the content of malondialdehyde (MDA) and activities of superoxide dismutase (SOD) and peroxidase (POD) increased in sweet corn, suggesting an activation of the reactive oxygen species (ROS) controlling pathways. Environmental stress can lead to overproduction of reactive oxygen species (ROS), which can cause oxidative damage to plants [7]. It is well known that most plants can resist oxidative damage through the ascorbate–glutathione (AsA-GSH) cycle in chloroplasts by increasing the activity of ascorbate peroxidase (APX, EC 1.11.1.11), monodehydroascorbate reductase (MDHAR, EC 1.6.5.4), dehydroascorbate reductase (DHAR, EC 1.5.5.1), and glutathione reductase (GR, EC 1.6.4.2) [8]. AsA-GSH cycle system can play an important role in maintaining protein stability, structural integrity of the biofilm system, and defense against membrane lipid peroxidation, such as nitric oxide (NO) induced by exogenous jasmonic acid (JA) upregulated the activity of the AsA-GSH cycle and had important role in drought tolerance [9]; the protective ability of exogenous triacontanol (TRIA) to modulate the redox status through the antioxidant pathway AsA-GSH cycle, so reducing Cd-induced oxidative stress [10]. Therefore, the AsA-GSH cycle is important for the protection against oxidative damage induced by stresses. 

Nitric oxide, a ubiquitous signal molecule, plays important roles in different plant tissues and participates in a variety of physiological processes. NO may also participate in the process of plant resistance to stress by regulating the production of endogenous hormones and increasing antioxidant enzyme activities in plants [11,12,13,14]. It has been reported that NO can participate in the activity of ABA-induced antioxidant enzymes, and can relieve the oxidative damage caused by high-intensity light stress in leaves of tall fescue [15]. Shan et al [16] demonstrated that NO regulated the AsA-GSH cycle in *Agropyron cristatum* under water stress. Previously, we studied the effect of ammonium nitrogen and nitrate nitrogen on seedlings of Chinese cabbage under low light stress. The results showed that supplying appropriate concentration of ammonium nitrogen could alleviate low light stress in the seedlings [17]. However, it is not clear what mechanism is involved in this alleviation. It is generally accepted that nitrite or L-arginine (L-Arg) is the main substrate for NO synthesis, both via enzymatic and non-enzymatic routes in plants [18]. The enzymatic pathway involves nitrate reductase (NR)/nitrite-NO reductase (Ni-NOR) or nonidentified nitric oxide synthase (NOS)-like enzyme [19]. The non-enzymatic pathway relies on nitrite reduction at acidic pH or through the action of the mitochondrial electron transport chain under anaerobic conditions [20]. Moreover, in hydroponic condition, the addition of ammonium in solution will affect the pH value of solution and nitrate metabolic pathway. Based on the NO synthesis in plants, we propose the hypothesis that NO participated in the regulation of the ascorbate–glutathione cycle induced by the appropriate ammonium: nitrate to mitigate low light stress. 

Mini Chinese cabbage (*Brassica pekinensis*), a subspecies of *Brassica rapa* belonging to the *Cruciferae*, is a mini-type Chinese cabbage. Because of its compact size, light texture, good taste, and high nutrient content, it is very popular among the people of China. In northwestern China, mini Chinese cabbage is widely cultivated during winter in solar-greenhouses where low light intensity (between 85 and 150 μmol m^−2^ s^−1^ in day) is a major abiotic stress factor limiting plant growth and crop productivity [21]. How to solve the problem of low light intensity stress on the growth and development of mini Chinese cabbage in China has become a priority for research. To address this, we propose the hypothesis that appropriate ammonium/nitrate ratio (10:90) can induce NO synthesis to regulate the AsA-GSH cycle in mini Chinese cabbage seedlings under low light intensity. The objectives of the experiment were (1) the effect of exogenous NO on the response of mini Chinese cabbage seedlings to low light stress; (2) to find out whether appropriate ammonium:nitrate ratio (10:90) can induce NO synthesis to regulate the AsA-GSH cycle in mini Chinese cabbage seedlings under low light intensity.

## 2. Results

### 2.1. Effects of NO and NH_4_^+^:NO_3_^−^ Ratio on Membrane Lipidation Damage in Leaf under Low Light Stress

Hydrogen peroxide (H_2_O_2_) and superoxide anion (O_2_^−^), malondialdehyde (MDA) contents in seedling are considered as important indices of evaluating the degree of membrane lipid damage to low light stress. As shown in Figure 1, under low light condition, the MDA contents in NH_4_^+^:NO_3_^−^ (10:90) and SNP treated leaf significantly decreased by 42.3% and 67.63% compared to the solo nitrate treated leaf, respectively. Whereas the addition of NO inhibitors (NaN_3_, L-NAME) and scavenger (Hb) in N treatment solution significantly increased the MDA contents compared with N treatment (Figure 1A).

Under low light stress, compared with the solo nitrate treatment, the appropriate NH_4_^+^:NO_3_^−^ ratio and exogenous NO treatments decreased H_2_O_2_ content by 42.5% and 57.34%, respectively and significantly reduced O_2_^−^ content by 42.3% and 52.54%, respectively. However, the treatment of N + NaN_3_, N + L-NAME, and N + Hb markedly increased the contents of H_2_O_2_ and O_2_^−^ (Figure 1B,C). The results suggested that the NO induced by appropriate NH_4_^+^:NO_3_^−^ ratio participates in the regulation of the membrane lipid oxidation damage in mini Chinese cabbage seedlings under low light stress.

### 2.2. Effects of NO and NH_4_^+^:NO_3_^−^ Ratio on the Contents of Redox State of Ascorbate and Glutathione in Leaf under Low Light Stress

GSH and AsA are two important antioxidants, which are two key components in the AsA-GSH cycle to maintain the cellular redox status. As shown in Figure 2A, the AsA contents in N (NH_4_^+^:NO_3_^−^ = 10:90) treated leaves was 96.3% higher than that in control (NH_4_^+^:NO_3_^−^ = 0:100) treated leaves. The exogenous NO (SNP) treatment was markedly increased AsA contents, being about 4-fold of the control. Compared with the N treatment, the addition of L-NAME, NaN_3_, and Hb in N solution decreased AsA contents, which reversed the effects of N on AsA content (Figure 2A). Similarly, the treatments of N and SNP were significantly increased the GSH contents by 66.1% and 93.3% compared with the control, respectively. The addition of NO inhibitors (NaN_3_, L-NAME) and scavenger (Hb) in N solution obviously decreased the GSH contents, as compared to the N treatment (Figure 2B). The data revealed that NO induced by appropriate NH_4_^+^:NO_3_^−^ ratio (10:90) plays an important role in increasing the AsA and GSH contents of mini Chinese cabbage seedlings under low light stress.

### 2.3. Effects of NO and NH_4_^+^:NO_3_^−^ Ratio on the Activities of Key Enzymes Involved in the AsA-GSH Cycle in Leaf under Low Light Stress

The interconversions between oxidized and reduced ascorbic and glutathione in leaves were also investigated. As shown in Figure 3, the appropriate NH_4_^+^:NO_3_^−^ and exogenous NO enhanced the activities of AAO, APX, DHAR, GR, and MDHAR in leaves compared with the solo nitrate treatment.

AAO is a key enzyme involved in the metabolism of substances in plants. It is a copper-containing enzyme that oxidizes ascorbic acid-producing water and dehydroascorbic acid (DHA). Compared with the control (NH_4_^+^:NO_3_^−^ = 0:100), the treatments of N and SNP were significantly increased the activities of AAO enzymes in leaves by 106% and 84.67%, respectively (Figure 3A). However, the activity of AAO was significantly decreased when the L-NAME, NaN_3_, and Hb were added in N treatment solutions separately.

APX is an important enzyme that scavenges excess of H_2_O_2_ by catalyzing it into water and divalent oxygen. As shown in Figure 3B, compared with control, the N and exogenous NO (SNP) treatment also markedly increased the activities of the APX enzymes with the values of 83.3% and 104%, respectively. However, the application of L-NAME, NaN_3_, and Hb in N treatment solution markedly decreased the activities of APX enzyme under low light stress, as compared to the treatment of N (Figure 3B).

In the process of plant resistance to stress, the enzymes of DHAR and GR are two key enzymes that remove ROS and maintain the stability contents of AsA and GSH in plants. As shown in Figure 3C,D, compared to the control, the activity of DHAR enzyme was obviously increased by 152.6% after the application of SNP, but when plants exposed to the treatments of N + NaN_3_, N+ L-NAME and N + Hb, the activity of DHAR decreased by 46.31%, 49.47%, and 47.36%, respectively (Figure 3C). The application of SNP slight increased the GR activity by 28.51%, as compared to the treatment of N, while the treatments of N + NaN_3_, N+ L-NAME, and N + Hb significant decreased the activity of GR enzymes when compared with the N treatment, and there is no significant difference between these treatments (Figure 3D). 

Compared with the control, the activity of MDHAR enzyme in leaves treated with N and exogenous NO was increased by 102% and 70.7%, respectively (Figure 3E). However, the application of L-NAME, NaN_3_, and Hb in N solution markedly decreased the MDHAR activity (Figure 3E). These results suggested that NO induced by appropriate NH_4_^+^:NO_3_^−^ ratio participated in the regulation of the key enzymes involved in AsA-GSH cycle under low light stress.

### 2.4. Effects of NO and NH_4_^+^:NO_3_^−^ Ratio on the Gene Expression Level Related to Antioxidative Enzymes in the AsA-GSH Cycle under Low Light Stress

To further verify whether NO induced by the appropriate NH_4_^+^:NO_3_^−^ ratio participates in the regulation of AsA-GSH cycle under low light stress, we used the RT-PCR technique to verify the relative expression levels of genes coded with the related enzymes which are involved in the AsA-GSH cycle. The relative expression level of *GR* gene in SNP-treated seedlings were significantly higher than that in solo nitrate-treated leaves, which increased by 38.72% (Figure 4A). Compared with the control, exogenous SNP treatment and N treatment slightly upregulated the relative gene expression level of *MDHAR1*, but the relative expression level of *MDHAR1* gene in N + NaN_3_, N + L-NAME, and N + Hb treated seedlings significantly downregulated when compared with N treatment (Figure 4B). From Figure 4C,D, we could see that the application of N and SNP were significantly upregulated the relative expression levels of *APXT and DHAR2* genes when compared with the control, which increased by 72.3% and 57.6%; and 57.7% and 56.9%, respectively. However, the application of L-NAME, NaN_3_, and Hb in N solution markedly downregulated the relative expression levels of the *APXT* and *DHAR2* genes (Figure 4C,D). The *AAO* gene expression level in N-and SNP-treated seedling were higher than that in solo nitrate-treated leaf, and increased by 20.6% and 8.4%, respectively (Figure 4E). The results from the transcriptional level verified that NO induced by appropriate NH_4_^+^:NO_3_^−^ ratio upregulated some vital genes coded with related enzymes involved in the AsA-GSH cycle. 

### 2.5. Effects of NO and NH_4_^+^:NO_3_^−^ Ratio on the NO Level in Root Tissues under Low Light Stress

The NO-dependent fluorescence intensity in root tips were detected to further prove that appropriate NH_4_^+^:NO_3_^−^ ratio induced NO synthesis and then participated in regulating AsA-GSH cycle under low light intensity. The levels of NO in root tips were determined according to the intensity of fluorescence. As shown in Figure 5, the levels of NO in root tips under N treatment were 38% higher than that under CK treatment. This suggests that appropriate NH_4_^+^:NO_3_^−^ ratio promoted the production of NO in root tips. Moreover, when Hb, NaN_3_, and L-NAME were applied in N solution, the levels of NO in root tips significantly decreased by 56.8%, 53.8%, and 32%, respectively, compared with N treatment. These data suggested that appropriate NH_4_^+^:NO_3_^−^ ratio increased biosynthesis of NO in tissues in mini Chinese cabbage under low light intensity. 

## 3. Discussions

Low light is one of the main factors that limit the productivity of many cultivated crops. However, accumulation of ROS such as O_2_^−^ and H_2_O_2_ occurred under low light condition was associated with signs of oxidative damage to lipids [7]. Malonaldehyde, which is a by-product of lipid peroxidation, is usually considered as an indicator of oxidative stress [22]. In previous studies, under 0 ℃ low temperature stress, the O_2_^−^ generation rate and H_2_O_2_ content increased rapidly in ginkgo leaf. Meanwhile, MDA content and membrane permeability increased remarkably, and the growth and development of the plant are hindered [23]. Membrane lipid oxidative damage occurred in *Vicia faba* seedlings under Cd stress, NO could reduce the contents of MDA, hydrogen peroxide, and reactive oxygen species (ROS) [24]. The appropriate ratio of ammonium and nitrate is more favorable for plant growth and development than the single application of ammonium nitrogen or nitrate nitrogen [25]. Our previous study showed that the application of appropriate NH_4_^+^:NO_3_^−^ (10:90; total nitrogen concentration 5 mmol/L) ratio could alleviate low light stress in mini Chinese cabbage seedlings [17]. In our present study, the accumulation of MDA, H_2_O_2_, and O_2_^−^ in appropriate N-and SNP-treated plants was lower than those in solo nitrate-treated plants, which alleviated the membrane lipid damage in plant and the alleviation effect of NO is more obvious than that of N under low light stress (Figure 1). That is to say, the same alleviation effects of N (appropriate NH_4_^+^:NO_3_^−^ ratio) on low light intensity was observed once again in the present study. Moreover, we found that exogenous NO could also decrease the contents of MDA, H_2_O_2_, and O_2_^−^ in mini Chinese cabbage under low light intensity, suggesting the exogenous NO had the similar effect with N on mitigating low light intensity stress. However, the alleviation effect of N on membrane lipid peroxidation was reversed by the addition of NO scavenger (Hb) and NO synthesis inhibitors (L-NAME and NaN_3_) in the N treatment solution (Figure 1). The NO level in root tissue under N treatment was significantly higher than that under nitrate only treatment (Figure 5). All these data suggested that the alleviation role of N on membrane lipid damage was due to the induction of endogenous NO synthesis. 

The ascorbate-glutathione (AsA-GSH) cycle is recognized to be a key player in H_2_O_2_ metabolism in higher plants and components of the cycle mainly located in plant cell, cytosol, mitochondria, and peroxisomes as well as the chloroplast [26]. AsA and GSH are the two important redox compounds of the antioxidant system and they can defense against stress-induced ROS to keep cellular redox homeostasis in plants [27,28,29]. Among the AsA-GSH cycle, there are four key enzymes (APX, GR, MDHAR, and DHAR) involved in interdependence and independence of AsA and GSH in H_2_O_2_ metabolism [26,30]. The AAO located in apoplast plays vital role in regulating the redox state of extracellular AsA pool, similar function with APX located in intracellular. Wu et al. reported that the value of oxidation/reduction ratio of AsA pool is regulated by AAO when cucumber seedlings were suffered from salt stress, and they deemed that it was the first defense of cells against stress damages [31]. As we known, the excess accumulation of H_2_O_2_ under abiotic stress will damage the membrane system and cell organelles of plants. Although AsA-GSH cycle is known as the antioxidant defense system in plant cell, its activity can also be regulated by environmental stresses [32,33,34]. In general, when plants were subjected to various stresses, the contents of GSH and AsA were changed and the activities of APX, GR, MDHAR, and DHAR were triggered [32,35]. Kanwar et al. [35] reported that arsenic toxicity treatments significantly enhanced the modulations of various stress markers like proteins, antioxidative enzymes (SOD, CAT, POD, APX, GR, MDHAR, and DHAR) and MDA content. Maintaining the efficient recycling of AsA via MDHAR and DHAR is crucial to maintaining AsA in a high redox state so that it can efficiently scavenge H_2_O_2_ via the catalytic reaction of APX and AAO [36]. In our present study, the exogenous NO and appropriate NH_4_^+^:NO_3_^−^ ratio increased the contents of AsA and GSH (Figure 2). Whereas the contents of AsA and GSH decreased significantly when the seedlings were treated with NO scavengers (Hb) and inhibitors (L-NAME, NaN_3_) in N solutions (Figure 2). Moreover, the NO level was significantly higher in appropriate NH_4_^+^:NO_3_^−^ ratio treated root than that in solo nitrate-treated root (Figure 5), indicating that NO induced by appropriate NH_4_^+^:NO_3_^−^ ratio participated in the antioxidant defense system under low light stress. Meanwhile, N and NO enhanced the activities of the key enzymes involved in GSH-ASA cycle under low light stress to maintain a favorable redox status (Figure 3). 

Similarly, it was noted that exogenous NO treatment effectively increased GSH content, effectively prevented wheat seedlings from being damaged under salt stress, and improved their resistance [37]. Based on our results, the activities of key enzymes involved in the AsA-GSH cycle and the contents of AsA and GSH under NO and N treatments were higher than those under the control treatment (Figure 2 and Figure 3). The scavengers and synthesis inhibitors of NO inhibited the production of endogenous NO in plants. The application of Hb, L-NAME, and NaN_3_ in N treatment solutions markedly decreased the activities of GR, AAO, MDHAR, DHAR, and APX enzymes under low light stress (Figure 3). It was suggested that NO involved in the regulation of the AsA-GSH cycle in mini Chinese cabbage seedling under low light stress. The results are similar to that reported by Shan et al. [9], who indicated that NO participated in the regulation of the AsA-GSH cycle in wheat seedlings under drought stress with the application of exogenous jasmonic acid. Similar results have also been reported in other stressed plants [38,39]. Begara-Morales et al. [40] demonstrated that ROS and NO interacted during plant responses to biotic/abiotic stresses and it was NO-related posttranslational modifications (PTMs), such as S-nitrosylation and/or tyrosine nitration, that regulated proteins linked to ROS metabolism. Moreover, they deemed that the enzymes involved in the AsA-GSH cycle were identified as NO targets using proteomic analytical techniques. The expression levels of *GR*, *MDHAR1*, and *APXT* genes were upregulated by N and SNP treatments, whereas they were downregulated by the N + Hb, N + L-NAME, and N + NaN_3_ treatments (Figure 4A–C). Similar phenomena were observed in the relative expression levels of *DHAR_2_* and *AAO* genes (Figure 4D,E). The upregulation of the *DHAR_2_* and *AAO* genes in NO-treated leaf was significant when compared with those in control-treated leaf and N + Hb, N + L-NAME, and N + NaN_3_ treated leaves (Figure 4). Previous research demonstrated that nitric oxide triggers multiple redox regulatory (defense-related) genes expression in stressed plants [41]. Sun et al. [39] reported that NO protects wheat root against Al-induced oxidative stress, possibly through the regulation of the AsA-GSH cycle, and NO application enhanced the related gene transcriptional level of AsA-GSH cycle in wheat genotypes, which is consistent with our results. The NO and N treatments upregulated the relative expression levels of *GR, MDHAR1, APXT, DHAR2*, and *AAO* genes and the application of L-NAME, NaN_3_, and Hb in N treatment solution markedly downregulated these gene expression levels. Moreover, the NO level in appropriate NH_4_^+^:NO_3_^−^ ratio treated root was significantly higher than that in solo nitrate-treated root (Figure 5), which suggesting that appropriate NH_4_^+^:NO_3_^−^ (10:90) mitigated low light intensity in mini Chinese cabbage seedling via inducing endogenous NO synthesis to regulate the AsA-GSH cycle.

In summary, we found that endogenous NO induced by appropriate NH_4_^+^:NO_3_^−^ (10:90) was involved in the regulation of ASA-GSH cycle in mini Chinese cabbage under low light intensity, through upregulating the relative expression level of key genes (*GR*, *MDHAR1*, *APXT*, *DHAR2*, and *AAO*), and then promoting the activities of related enzymes (GR, MDHAR, APX, DHAR, AAO) to scavenge H_2_O_2_ and O_2_^−^. Decreasing the accumulation of ROS and exogenous NO also played an important role in the process. Finally, we concluded that appropriate NH_4_^+^:NO_3_^−^ (10:90) induced NO synthesis which played an important role in regulating the AsA-GSH cycle in mini Chinese cabbage seedlings under low light stress.

## 4. Materials and Methods

### 4.1. Plant Material and Growth Conditions

Mini Chinese cabbage (*Brassica pekinensis* cv. “Jinwa no. 2”) seeds were purchased from Gansu Agricultural Institute (Lanzhou, China). The seeds were surface-sterilized in 5% sodium hypochlorite for 10 min, then washed with water. The seeds were soaked in water for 6 h then placed in petri dishes with double-layer filter paper moistened with distilled water, and finally kept in the dark to germinate at 25 °C for 16 h. After germination, uniformly geminated seeds were sown in pots filled with clean quartz sand and cultured with half-strength whole Hoagland’s nutrient solution for 10 days, with a photoperiod of 12 h, temperature of 23 ± 2/13 ± 2 °C (day/night), and light fluence about 200 μmol m^−2^·s^−1^, in a modern climate controlled greenhouse.

### 4.2. Treatments and Experimental Design

At the two-leaf stage, uniform seedlings were transferred into distilled water for 1 d, after that, groups of four uniform seedlings were transplanted into a black circular container (diameter 12 cm, height 4.5 cm) filled with 250 mL the following treatment liquid for cultivation: CK (NH_4_^+^:NO_3_^−^ = 0:100, total nitrogen concentration 5 mmol/L); N (NH_4_^+^:NO_3_^−^ = 10:90,total nitrogen concentration 5 mmol/L); SNP (NO donor, sodium nitroprusside, 100 μmol/L); N + Hb (NO scavenger, hemoglobin, 0.1%); N + L-NAME (NOS inhibitor, N-nitro-l-arginine methyl ester, 25 μmol/L); N + NaN_3_ (NR inhibitor, sodium azide, 50 μmol/L). Each treatment had six replications. 7 μmol/L nitrification inhibitor dicyandiamide (C_2_H_4_N_4_) was added to each pot of treatment solution, which inhibited the conversion of nitrate nitrogen to ammonium nitrogen. The pH of the treatment solution was adjusted to 6.5–7.0 with 0.1 mol·L^−1^ hydrochloric acid or sodium hydroxide. The seedlings cultivated in black containers filled with different solutions were placed in a light incubator, the light intensity was 100 μmol·m^−2^·s^−1^ (moderately low light stress), the photoperiod was 12 h/12 h, and the temperature was 25 ± 1°C/18 ± 1 °C (day/night), and the humidity is 60%. In order to avoid root hypoxia and lack of nutrient solution, replace the treatment solution once every 2 days. After 7 days of treatment, measurement of various indicators began.

### 4.3. Determination of the Contents of Hydrogen Peroxide (H_2_O_2_), Superoxide Anion Free radical (O_2_^−^), and Malondialdehyde (MDA) 

Sample of leaf tissue (0.5 g) was homogenized in 0.1% (w/v) cold trichloroacetic acid (TCA). After centrifugation at 12,000 *g* for 20 min, the extracted supernatant was used for determination of H_2_O_2_, O_2_^−^, and MDA content following the method of Alexieva et al. [42] with a few modifications. The reaction mixture contained 0.5 mL extracted supernatant, 0.5 mL 100 mM phosphate buffer (pH 6.8) and 2 mL reagent (1 M KI w/v in fresh double-distilled water). 0.1% TCA solution without material extract was used as the blank. The mixture was incubated for 1 h in the dark and the absorbance was measured at 390 nm. The content of hydrogen peroxide was calculated according to a standard curve. 

To determine the content of MDA, the reaction mixture contained 0.5 mL extracted supernatant and 1 mL 0.5% (w/v) thiobarbituric acid (TBA) in 20% TCA. The mixture was incubated at 100 °C for 30 min, and the samples were centrifuged at 10,000× *g* for 10 min, and the absorbance of supernatants was measured at 532 nm. The value for non-specific absorption at 600 nm was subtracted.

The O_2_^−^ level was analyzed according to the method of Nakajima et al. [43] with some modifications, the reaction mixture contained 2 mL extracted supernatant, 1.5 mL phosphate buffer, and 0.5 mL hydroperamine hydrochloride. The mixture was incubated for 20 min in 25 °C water bath. The 2 mL the above reaction solution was mixed with 2 mL 17 mM aminobenzene sulfonic acid for 30 °C constant temperature reaction for 30 min, and the absorbance of supernatants was measured at 530 nm.

### 4.4. Determination of Glutathione (GSH) and Ascorbate (AsA)

GSH and AsA contents of mini Chinese cabbage leaves were measured using the GSH and AsA determination Kit (Beijing Solarbio Science & Technology Co., Beijing, China) following the manufacturer’s instructions. 

### 4.5. Determination of Activities of Enzymes of AsA-GSH Cycle

The activities of ascorbate oxidase (AAO), ascorbate peroxidase (APX), glutathione reductase (GR), monodehydroascorbate reductase (MDHAR), and dehydroascorbate reductase (DHAR) in leaves were analyzed using the AAO, APX, GR, DHAR, and MDHAR determination Kit (Beijing Solarbio Science & Technology Co., Beijing, China) according to the manufacturer’s instructions.

### 4.6. Transcript Level Estimation with RT-PCR

Total RNA was isolated from 100 mg (fresh weight) of excised mini Chinese cabbage seedling leaves by grinding with mortar and pestle in liquid nitrogen to obtain a fin paste using TaKaRa reagent (TaKaRa Bio, Japan) according to the manufacturer’s instructions. Each treatment was replicated three times. Primers designed for the genes and reference genes are detailed in Table 1. Each reaction (20 μL total volume) consisted of 10 μL SYBR Premix Ex Taq II, 2 μL of diluted cDNA and 0.8 μL of forward and reserve primers. PCR amplification conditions were as follows: 1 min at 95 °C followed by 40 cycles of 5s at 95 °C and 20 s at 60 °C with data collection at the annealing step. After the 40 cycles, the dissociation/melting curve stage with 0 s at 95 °C, 15 s at 65 °C, 0 s at 95 °C and 30 s at 50 °C was included. The relative quantification of mRNA levels is based on the method of Livak and Schmittgen [44]. The expression level relative to the control for each sample was expressed as 2^−ΔΔ*C*t^.

### 4.7. Detection of NO Level in Root Tissues with Fluorescence Probe 4,5-Diaminofluorescein Diacetate (DAF-2DA)

The NO-dependent fluorescence in root tissues was detected with the fluorescence indicator 4, 5-diaminofluorescein diacetate (DAF-2 DA; Sigma, Ronkonkoma, NY, USA) which can permeate into the cell membrane and emit fluorescence upon binding to NO. The root samples were incubated for 2 h under dark condition with 20 μM DAF-2DA in 50 μM Tris-HCl buffer (pH 7.5), and washed three times in fresh buffer. These were then visualized under a fluorescence microscope (Leica 400×, Planapo, Wetzlar, Germany) with excitation and emission wavelengths of 485 and 538 nm, respectively. The fluorescence intensity of fluorescent image was determined by ZEN2 software (Carl Zeiss). The fluorescence intensity, called ‘relative fluorescence unit’, was expressed in color level on a scale ranging from 0 to 255.

### 4.8. Statistical Analysis

The analysis of variance was performed using SPSS Statistics 17.0 software. The significance of means difference between treatments was tested by Duncan’s multiple tests (*p* < 0.05).

## Figures and Tables

**Figure 1 plants-08-00489-f001:**
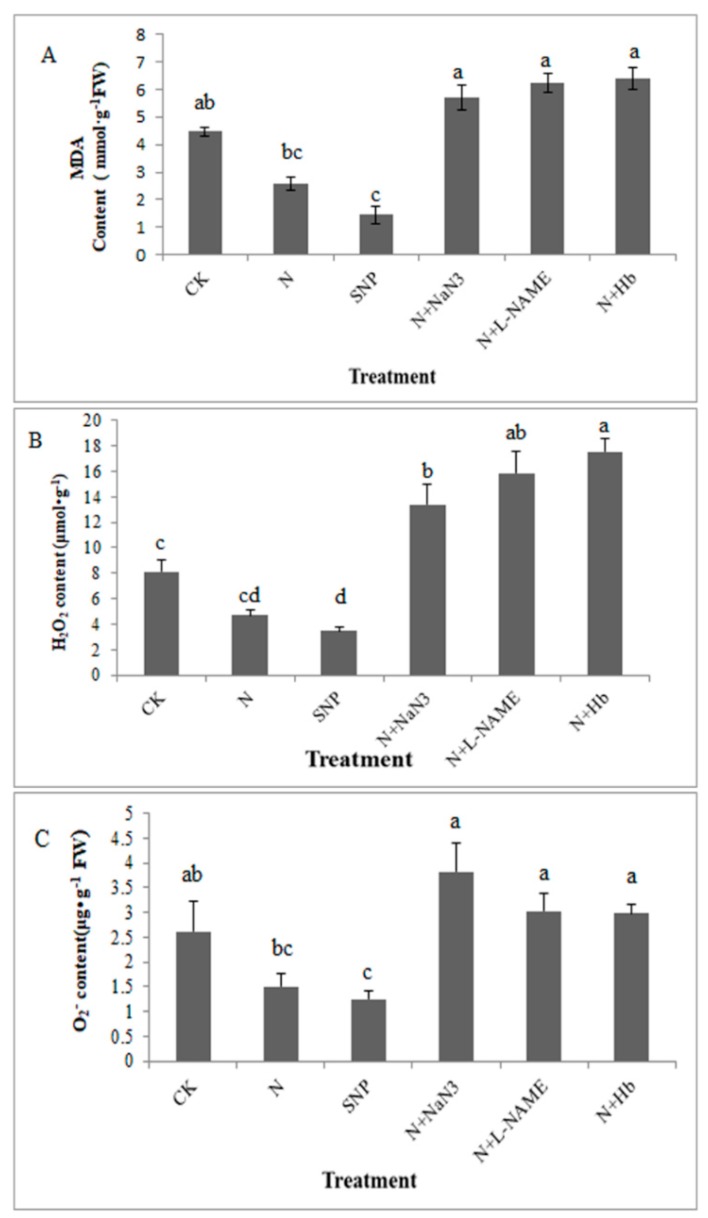
Effects of NO and NH_4_^+^:NO_3_^−^ ratio on membrane lipidation damage in mini Chinese cabbage leaves under low light stress. The seedlings cultivated in black containers filled with different solutions, CK (NH_4_^+^:NO_3_^−^ = 0:100); N (NH_4_^+^:NO_3_^−^ = 10:90); SNP (100 μmol/L); N + Hb (0.1%); N + L-NAME (25 μmol/L); N + NaN_3_ (50 μmol/L) were treated for 7 days. Capital letters (**A**–**C**) represent MDA content, H_2_O_2_ content, O_2_^−^ content, respectively. Bars indicated the SE (n = 3). Significant differences (*p* < 0.05) between treatments are indicated by different letters, for low light condition.

**Figure 2 plants-08-00489-f002:**
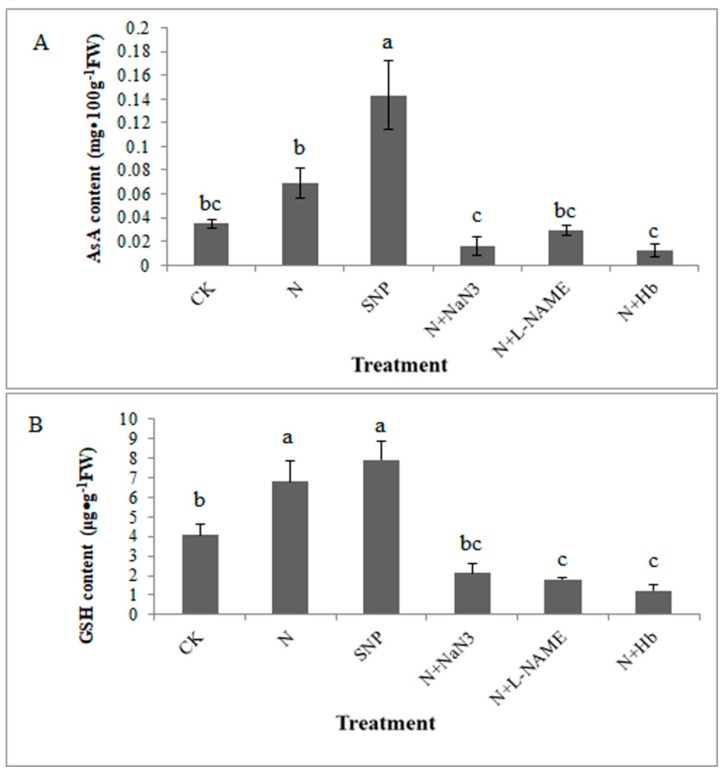
Effects of NO and NH_4_^+^:NO_3_^−^ ratio on the contents of redox state of ascorbate and glutathione in mini Chinese cabbage leaves under low light stress. The seedlings cultivated in black containers filled with different solutions, CK (NH_4_^+^:NO_3_^−^ = 0:100); N (NH_4_^+^:NO_3_^−^ = 10:90); SNP (100 μmol/L); N + Hb (0.1%); N + L-NAME (25 μmol/L); N + NaN_3_ (50 μmol/L) were treated for 7 days. Then the contents of AsA (**A**) and GSH (**B**) from mini Chinese cabbage leaves were analyzed. Bars indicated the SE (n = 3). Significant differences (*p* < 0.05) between treatments are indicated by different letters, for low light condition.

**Figure 3 plants-08-00489-f003:**
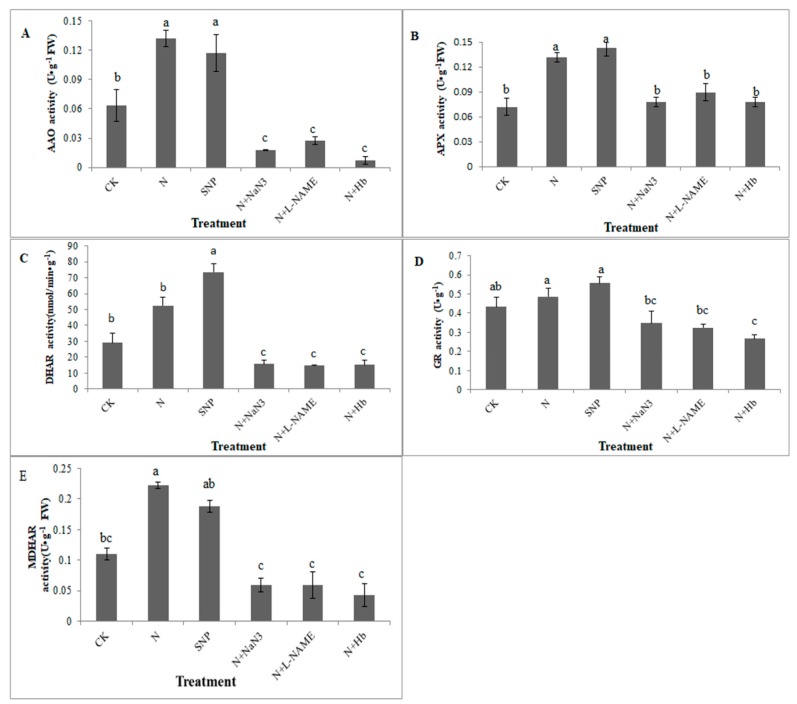
Effects of NO and NH_4_^+^:NO_3_^−^ ratio on the activities of key enzymes involved in the AsA-GSH cycle in mini Chinese cabbage leaves under low light stress. The seedlings were exposed to the different treatment solutions as described in Figure 1 for 7 d. Then the activities of AAO (**A**), APX (**B**), GR (**C**), DHAR (**D**), and MDHAR (**E**) from mini Chinese cabbage leaves was analyzed. Values represent mean ± standard error (SE) of three replications for each treatment, different letters indicated significant difference at *p* < 0.05 according to Duncan’s test.

**Figure 4 plants-08-00489-f004:**
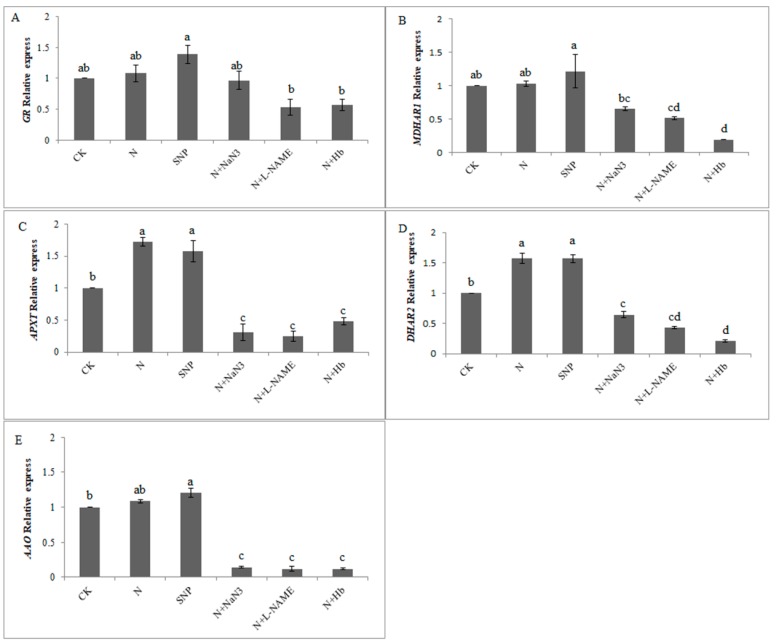
Effects of NO and NH_4_^+^:NO_3_^−^ ratio on the relative expression level of genes related to ant oxidative enzymes in the AsA-GSH cycle in leaf under low light stress. The seedlings were exposed to the different treatment solutions as described in Figure 1 for 7 d. Capital letters (**A**–**E**) represent the relative expression levels of *GR*, *MDHAR1*, *APXT*, *DHAR2*, and *AAO*, respectively. Data are the means of three replicates with SE shown by vertical bars. Means followed by the different letters are significantly different according to Duncan’s test (*p* < 0.05), for each treated condition.

**Figure 5 plants-08-00489-f005:**
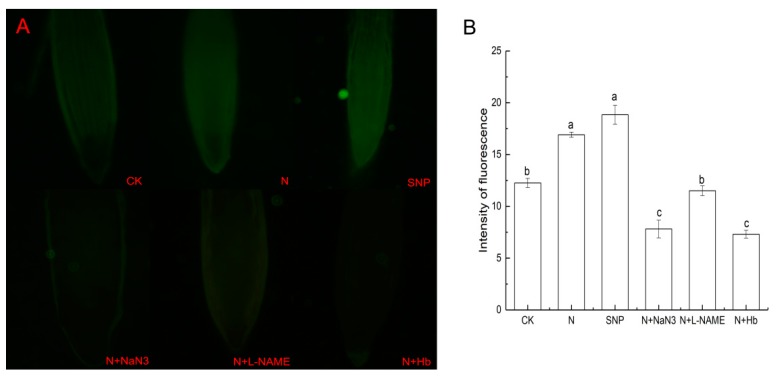
Effects of NO and NH_4_^+^:NO_3_^−^ ratio on the NO level in root tissues under low light stress. (**A**) Representative fluorescence images of DAF-2 DA-loaded roots treated for 7 days. (**B**) Pixel intensity of fluorescence in tips of the root treated for 7 days. The data are mean ± SE (n = 3). Different letters indicate significant differences (*p* < 0.05) between treatments. CK, NH_4_^+^:NO_3_^−^ = 0:100; N, NH_4_^+^:NO_3_^−^ = 10:90; SNP, 100 μmol/L; N + NaN_3_, 50 μmol/L NaN_3_ was added in N solution; N + L-NAME, 25 μmol/L. L-NAME was added in N solution; N + Hb, 0.1% Hb was added in N solution.

**Table 1 plants-08-00489-t001:** Gene specific primers used for real-time PCR.

Genes	Accession Number	Primer Name	Primer Sequence
*AAO*	Bra002355	Primer F	TGATGCTACCGCCGGAGACAC
		Primer R	TGCCGTGCCAATGGATGACAAC
*MDHAR1*	Bra006954	Primer F	GGCGGTGGCTCCTTATGAACG
		Primer R	TCCACCACTACCAACACAGCAATG
*DHAR2*	Bra008188	Primer F	TCCTCCTGAGTTCGCCTCTGTTG
		Primer R	GCCTTGTCGGAACCGTCACTG
*APXT*	Bra015668	Primer F	TCGCCTCCTCCTCCTCCTCTC
		Primer R	ACCACCGTGTTACTAGAGCCTCTG
*GR*	Bra001931	Primer F	GCTGGAGCTGTGAAGGTTGATGAG
		Primer R	CCATTAAGGCAACAGGCGTGAGG
*Actin*	JN120480.1	Primer F	CCAGGAATCGCTGACCGTAT
		Primer R	CTGTTGGAAAGTGCTGAGGGA

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
