# Peer review of "Nitric Oxide Is Involved in the Regulation of the Ascorbate–Glutathione Cycle Induced by the Appropriate Ammonium: Nitrate to Mitigate Low Light Stress in Brassica pekinensis"

_plants, 2019, doi:10.3390/plants8110489_

Round 1

Reviewer 1 Report

Dear Authors

I recomend you to write the Introduction and Discussion de novo after careful studies of et least 2 review papers:

 Foyer CH, Noctor G (2011) Ascorbate and Glutathione: The Heart of the Redox Hub. Plant Physiology 155: 2-18. http://www.plantphysiol.org/content/155/151/152.full Besson-Bard A, Pugin A, Wendehenne D (2008) New insights into nitric oxide signaling in plants. Annu Rev Plant Biol 59: 21-39; https://www.researchgate.net/publication/5816674_New_Insights_into_Nitric_Oxide_Signaling_in_Plants

At present there is a lot of misstatements in description of NO appearing in tissues, functions and detoxifications. I could not found the explanations what are the relation between different nitrogen forms in plant tissues.

Also a brief information about the nitrogene supplementation in aqua-cultures would be interesting.

I suppose that light intensity 200 micromoles is typical for winter, when the cabbage is grown in greenhouses. Such information should be given in the Introduction

Reviewer 2 Report

Nitric oxide (NO) has long been shown to regulate glutathione synthesis by upregulating gene expressions of gamma-glutamylcysteine synthetase (γ-ECS), glutathione synthetase (GSHS) (Innocenti et al. Planta 2007). In the submitted manuscript, authors try to connect the alleviation of low light stress on plant from application of ammonium/nitrate (10:90) with NO regulating glutathione synthesis. Although, by treating Chinese cabbage with exogenous NO donor or NO scavenger, inhibitors of NO synthesis, authors demonstrated involvement of NO signaling in regulating activity of ascorbate-glutathione cycle (basically repeat previous research), in order to put application of ammonium/nitrate (10:90) in the context, key evidence is missing, which is whether application of ammonium/nitrate (10:90) increases biosynthesis of NO in tissues. I suggest fluorescence probe diaminofluorescein (DAF) assay should be used to detect the changes of NO level in the root tissues with of ammonium/nitrate (10:90) compared with nitrate only.

Round 2

Reviewer 1 Report

Dear Authors

corrections which were done are satisfactory, so I recommend the paper for publishing. Congratulations

Krystyna Rybka

Reviewer 2 Report

I appreciate improvements that have been made. Quantification of native NO really provides solid supports for the final conclusion.